

# The effect of elevational gradient on alpine gingers (*Roscoea alpina* and *R. purpurea*) in the Himalayas

Babu Ram Paudel[1,2,3], Adrian G. Dyer[4], Jair E. Garcia[4] and Mani Shrestha[4]

[1] Yunnan Key Laboratory of Plant Reproductive Adaption and Evolutionary Ecology, Yunnan University, Kunming, Yunnan, China
[2] Laboratory of Ecology and Evolutionary Biology, State Key Laboratory for Conservation and Utilization of Bio-Resources in Yunnan, Yunnan University, Kunming, Yunnan, China
[3] Department of Botany, Prithvi Narayan Campus, Tribhuvan University, Pokhara, Gandaki, Nepal
[4] School of Media and Communication, RMIT University, Melbourne, Victoria, Australia

## ABSTRACT

There is currently enormous interest in how morphological and physiological responses of herbaceous plants may be affected by changing elevational gradient. Mountain regions provide an excellent opportunity to understand how closely related species may adapt to the conditions that rapidly change with elevation. We investigated the morphological and physiological responses of two Himalayan alpine gingers (*Roscoea alpina* and *R. purpurea*) along two different vertical transects of 400 m, *R. purpurea* between 2,174–2,574 m a.s.l and *R. alpina* between 2,675–3,079 m a.s.l. We measured the variables of plant height, leaf length, leaf area, specific leaf area, and stomata density at five plots, along the vertical transect at an elevational gap of ca. 100 m. Results revealed that with increased elevation plant height, and leaf area decreased while stomata density increased, whereas changes in specific leaf area, were not correlated with the elevation. Our results reveal that these alpine gingers undergo local adaptation by modifying their plant height, leaf area and stomata density in response to the varying selection pressure associated with the elevational gradient. Thus, the findings of this research provide valuable information on how a narrow range of elevational gradient affects the herbaceous plants at the alpine habitat of the Himalayas.

## INTRODUCTION

The elevational gradient is one of the key environmental factors that affect growth, morphology and physiology of plants (*Cordell et al., 1998*; *Hultine & Marshall, 2000*; *Qiang et al., 2003*). The elevational gradient in alpine regions provides a sharp environmental change across relatively short spatial distances because small changes in elevation can lead to a large shift in temperature, humidity, exposure, and concentration of atmospheric gases (*Hovenden & Vander Schoor, 2004*). Thus, alpine environments can provide useful natural avenues to investigate the response of plants to a suite of climatic conditions that are representative of the broader latitudinal range (*Montesinos-Navarro et al., 2011*). With the increase in elevation, there is typically an increase in both precipitation and light

Corresponding author
Babu Ram Paudel, babu@xtbg.ac.cn, brp2033@gmail.com

intensity including changes in distributions of short wavelength UV-A (315–400 nm) and UV-B (280–315 nm) radiation (*Diffey, 1991*; *Rozema et al., 1997*) whilst temperature and concentration of carbon dioxide and oxygen decrease (*Friend & Woodward, 1990*). These environmental variations may potentially alter the morphology and physiology of plants to endure the different stresses linked with changing elevation (*Hovenden & Brodribb, 2000*; *Körner, 2007*).

The alpine environment is potentially affected by climate change associated with global warming, and thus alpine plants may face rapidly changing environmental conditions that likely impose different stress levels on plants (*Beniston, 2003*; *Byars, Papst & Hoffmann, 2007*). Thus, based on the adaptive plasticity, the plant species exhibit local adaptation by altering the morphological and/or physiological traits over the range of elevational gradient (*Hirano, Sakaguchi & Takahashi, 2017*). For example, local adaptations of plants in response to variable climatic conditions at different elevations may result in variation of plant height and leaf length (*Wang & Gao, 2004*). Variation in carbon assimilation, energy balance and water relations along the elevational gradient could result in variation of leaf morphological and physiological traits such as leaf area, specific leaf area (SLA) and stomata density (*Ackerly et al., 2002*). Therefore, the study of the variation in the growth forms, morphology and physiology of a plant species along an elevational gradient could provide valuable insights on how plants may respond to environmental stress imposed by rapid changes in climatic conditions (*Premoli & Brewer, 2007*; *Körner, 2007*; *Bresson et al., 2011*).

Although several previous studies have documented the effects of elevational gradient on the growth, morphology and physiology of the plants, most of the studies are focused on tree species (*Cordell et al., 1998*; *Hultine & Marshall, 2000*; *Li et al., 2008*). Recently a few studies have been conducted to understand how the elevational gradients affect the herbaceous plants (*Gonzalo-Turpin & Hazard, 2009*; *Scheepens, Frei & Stöcklin, 2010*; *Hulshof et al., 2013*; *Bastida, Rey & Alcántara, 2015*; *Takahashi & Matsuki, 2017*; *Kiełtyk, 2018*). These studies particularly focused on the variation of a specific trait, such as vegetative trait, reproductive trait or leaf trait. Currently however, there is a lack of empirical evidence on the adaptive potential of herbaceous plants along the elevational gradient in steep environments such as the Himalayas. As the alpine ecosystem in the Himalayas is likely to experience the adverse effects of the changing climate associated with global warming and anthropogenic disturbances (*Beniston, 2003*; *Byars, Papst & Hoffmann, 2007*), understanding the performance of herbaceous plants along the elevational gradient provides important insights for the enhanced prediction of the response of herbaceous plants under altered climatic conditions.

The genus *Roscoea*, with 22 known species, is a Himalayan endemic alpine perennial herb and the only alpine member of the predominately-tropical family Zingiberaceae (*Cowley, 1982*; *Cowley, 2007*). The genus is distributed between the elevations of ca 1,500 to 4,500 m a.s.l (*Cowley, 2007*), thus serving as a key model for how herbaceous plants respond to the potentially stressful environmental conditions associated with increasing elevation. All *Roscoea* species are small herbs with annual leafy shoots produced from a reduced erect rhizome (*Cowley, 1982*; *Cowley, 2007*). Among the *Roscoea* species, *R. alpina*

Royle and *R. purpurea* Smith are widely distributed in the Himalayan Mountains from Kashmir (Pakistan) in the west through Nepal, India, Bhutan and Tibet. As these two *Roscoea* species are widely distributed from low to high elevations, characterization of the variation in morphological and physiological traits along the elevational gradient will help to understand how these alpine gingers respond to changes in climatic conditions associated with elevation. In this study, we explore the changes in the morphological (plant height and leaf length) and physiological variables (leaf area, SLA and stomata density) of these alpine gingers along the well-defined elevational gradient in the Himalayan mountain range.

## MATERIALS AND METHODS

### Study species

The two widespread *Roscoea* species used in this study were *R. alpina* and *R. purpurea* (Fig. 1). *Roscoea alpina* is a common species with a wide distribution between the elevations 2,130–4,270 m a.s.l in the Himalayan range from Kashmir (Pakistan) in the west through Bhutan in the east. The annual pseudostem may grow up to 12–20 cm high and presents flowers from the end of May to mid-August (*Cowley, 2007*). It has 2–3 obtuse sheathing leaves. Leaves are usually 1–2 in number and underdeveloped; occasionally the plant may bear up to four well-developed leaves. Leaves are linear, broadly elliptic or lanceolate. Only the first leaf is slightly auriculate and widest at the base while rest of the leaves are widest at the middle, with 17–25 cm in length. Leaves are usually glabrous but young leaves are occasionally hairy at acute apex. Inflorescences are without exserted peduncle. Flowers are deep purple to white in appearance for a human observer (Fig. 1). A single plant can develop up to five flowers, however only one flower blooms at a time. Obtuse to almost truncate bracts are shorter than the ovary. The calyx is much longer than the bract and bluntly bi-dentate. A long corolla tube is exserted from the calyx (*Cowley, 2007*).

*Roscoea purpurea* is also a widespread member of the Himalayan *Roscoea*, distributed between the elevations 1,520–3,100 m a.s.l. from Himachal Pradesh (India) in the west through Assam/Bhutan in the east. The annual erect pseudostem is most variable in habit and form and may grow up to 25–38 cm high, bearing 0–2 obtuse to truncate sheathing leaves. Leaves are usually 4–8 in number, lanceolate to oblong-ovate and 14–20 cm long with acuminate and sometimes with ciliated apex. Lower leaves are slightly auriculate at the base. The plant flowers from the end of June to early September (*Cowley, 2007*). The inflorescence is enclosed in upper leaf sheaths with only the upper part of bracts and flowers visible. Flowers are light purple or white with purple markings. Usually, 1–2 flowers open at a time. Bracts longer than calyx with acute apex which is pale green. The sharply bi-dentate and apiculate calyx is usually pale green and sometimes marked with pink. The corolla tube has a mauve or white colouration and is hardly exserted from the calyx (*Cowley, 2007*).

### Study sites

The research was conducted along an elevational gradient at two sites, Daman and Ghorepani, Central Nepal (Fig. 2). The Department of Plant Resources, Thapathali, Kathmandu provided research permission (125/05-19). Daman is located in Makawanpur

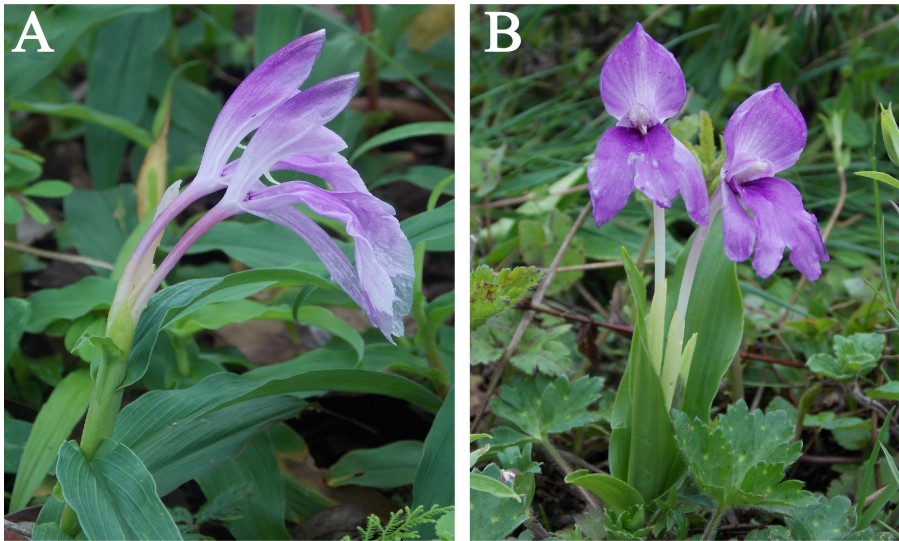

**Figure 1** Study species *Roscoea purpurea* (A) and *R. alpina* (B) in their natural habitat.

district and forms a part of the Mahabharat mountain range (mountains lower than the Himalayas). This site lies about 70 km south-west of Kathmandu and is midway between Kathmandu and Hetauda. The vegetation type of this site typically comprises a mixed forest of *Pinus* (*Pinus roxburghii*), *Rhododendron* (*R. arboreuum, R. campanualatum*) and *Quercus (Q. semecarpifolia, Q. lanata)*. The site experiences cool temperate to subalpine climate with warm summers and cold winters that typically incur mild to heavy snowfall from November to February (BR Paudel, Pers. Obs., 2014 and BR Paudel, Pers. Obs., 2017). Ghorepani, located in Myagdi district, is about 270 km west of Kathmandu. The vegetation type of this site comprises a mixed forest of *Pinus (P. wallichiana), Abies (A. spectabilis)* and *Rhododendron (R. arboretum, R. barbatum, R campanulatum, R antohopogon* at upper limit). The site has a subalpine climate and cool weather throughout the year, and heavy snowfall from November to February (BR Paudel, Pers. Obs., 2014 and BR Paudel, Pers. Obs., 2017). The geographical coordinates and the elevations of the study sites are presented in Table 1.

## Measurement of traits

The field sampling was conducted from May to August 2014 and repeated the sampling again in 2017 (May to August). Department of Plant Resources, Thapathali, Kathmandu, Nepal Approval number:125/05-19. Five sampling plots were selected along a vertical transect from 2,174 to 2,574 m a.s.l. for *R. purpurea* and from 2,675 to 3,079 m a.s.l. for *R. alpina*. The sampling was done in a counterbalanced random fashion such that two adjacent sampling plots were at an elevation gap of ca 100 m. Plant height and leaf length were measured to examine the morphological variables. Physiological variables included leaf area, specific leaf area (SLA) and stomata density. At each sampling plot, a horizontal transect of 100 m length was laid down and twenty plants were randomly selected along the horizontal transect in such a way that the distance between the adjacent sampling plant was
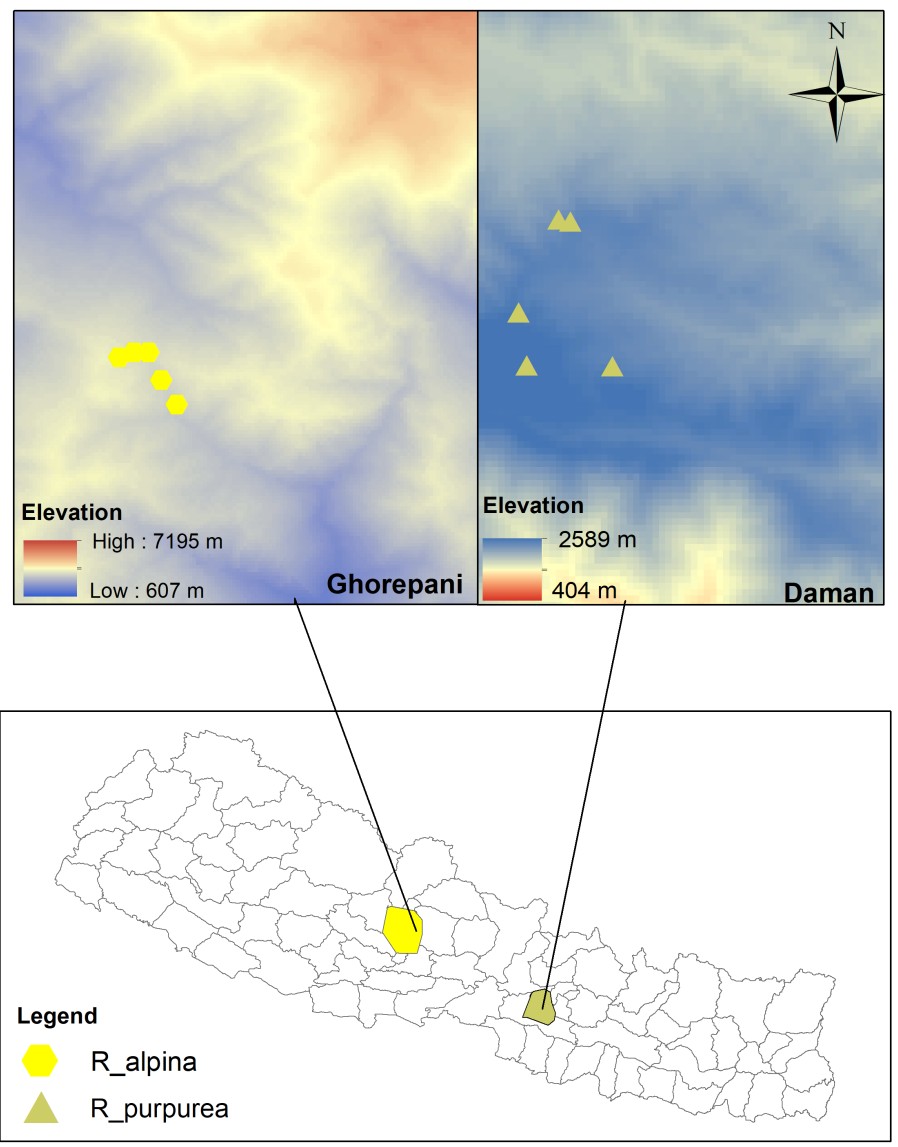

**Figure 2** **Map of the study area.** Yellow hexagons represent the study site (Ghorepani) of Roscoea alpina whereas Light-Oliventine triangles represent the study site (Daman) of R. purpurea (See Table 1 for detail). The top map represents the elevational gradient of study locations.

at least 5 m. A standard metric ruler was used to measure plant height (the distance from the ground to the topmost part of the stem). The largest leaf of every sampled plant was removed and leaf length was measured with a ruler. We used a graph paper to trace and quantify the area of each leaf, enabling robust repeatable measurements in remote locations. Specifically, two alternative methods were used to measure the area of the leaf. In 2014, the area of the leaf was measured after wet storage, while in 2017 the area was measured on the freshly plucked leaf. To prevent the leaves from possible shrinkage during wet storage, the leaves were first flattened if necessary and carefully placed in between the folds of a paper.

**Table 1 Geographical details of study sites.**

| | *Roscoea purpurea* | | | *R. alpina* | |
|---|---|---|---|---|---|
| Latitude | Longitude | Elevation (a.s.l.) | Latitude | Longitude | Elevation (a.s.l.) |
| 27°36′45.7″N | 85°5′32″E | 2,174 m | 28°23′21.9″N | 83°42′22.1″E | 2,675 m |
| 27°36′44.7″N | 85°5′37.6″E | 2,274 m | 28°23′42.2″N | 83°42′9.2″E | 2,770 m |
| 27°36′2.1″N | 85°5′13.4″E | 2,374 m | 28°24′4.9″N | 83°41′58.9″E | 2,874 m |
| 27°35′37.1″N | 85°5′57.3″E | 2,474 m | 28°24′5.2″N | 83°41′46.8″E | 2,968 m |
| 27°35′37.4″N | 85°5′17.3″E | 2,574 m | 28°24′0.9″N | 83°41′34.7″E | 3,079 m |

Resulting samples were then placed in a sample box to avoid the external light and heat sources. For both respective leaf collection methods, the leaf (either wet stored or freshly plucked) was placed on a graph paper, its outline was sketched and the number of squares enclosed within the leaf-outline were counted. Complete and greater than half squares were scored, whilst squares less than half a square were excluded. The measurements were repeated several times for each leaf to enable a robust field measurement of leaf area. Twenty leaves at each sampling plot were measured to assess variability. The area of leaf as measured by two alternative methods did not differ significantly ($t$ test, $P > 0.05$), thus data generated from the freshly removed leaf were used for further analysis. All collected leaves were gently pressed between the folds of an absorbent paper for five days to flatten the leaf surface and to absorb any excess moisture. The pressed leaves were subsequently oven dried at the university laboratory for 48 h at 70 °C. Dry leaf weight was measured using a digital electronic balance (Fameway International (HK) Limited; accuracy 0.001 g). Specific leaf area (SLA) of a leaf was calculated as the ratio of the area of a fresh leaf and its dry weight and expressed in cm$^2$/g.

To determine the stomatal count, transparent nail polish was applied on the middle dorsal surface of a fresh leaf. After a few minutes, when nail polish had dried, a thin layer was peeled from the middle dorsal surface of a leaf. The peeled layers were separately preserved in a 10% glycerine solution for about 72 h. In the laboratory, the temporary slide of each layer was prepared using safranin as a staining agent. The stained layers were individually mounted on microscope slides, and all stomata observed under a 10-x magnification microscopic field were counted. The stomata counts were repeated at three different microscopic fields to ensure the exact measurement of the stomata density. Area of the microscopic field was calculated using the formula $A = \pi r^2$ where r is the radius of microscopic field and density of stomata was calculated as the number of stomata under a microscopic field divided by the area of the microscopic field. The stomata density was expressed in terms of number per square millimetre.

## Statistical Analyses

An independent sample $t$ test was used to test the variation in measured traits between the years. Data from each of the measured morphological variables were summarized as Q-Q plots and tested for normality. Exploratory data analyses revealed that some of the response variables were not normally distributed and were better described by a Gamma

distribution as most data consisted of positive values larger than zero (*Zuur, Hilbe & Ieno, 2013*). Consequently, non-parametric correlation analyses were performed among the five different traits measured for each species implementing Kendall's tau statistic ($\tau$). This coefficient was chosen as it has a known standard error and provides a better estimate with low sample size. After the exploratory analyses, generalised linear regression models (GLM) were applied to test for the potential effects of elevation on the different traits measured for each species. For the five regression models, elevation was used as a predictor and it was assumed that the response variable followed a Gamma distribution. Link function for each model was selected based on a comparison of AIC scores obtained after fitting models implementing different link functions (*Zuur, Hilbe & Ieno, 2013*). Regression analyses were performed using the routine *glm* available as part of the base distribution of the R package (version 3.3.1) (*R Core Team, 2015*).

## RESULTS

### Correlation analyses

Our results indicated that all the measured variables did not differ significantly between years ($P > 0.05$), thus only 2017 data were used for further analyses. For *R. purpurea*, there was a significant correlation between stomata density and the variables of leaf area ($P < 0.001$) and specific leaf area (SLA) ($P = 0.020$) (Fig. 3). In *R. alpina*, leaf length was correlated with all remaining variables (Fig. 4). Consequently, we separately performed the regression analyses for the two species for each of the measured response variables.

### Variations of traits with elevation

Leaf length of *R. alpina* and plant height significantly decreased with increasing elevation ($P = 0.001$ for leaf length and $P = 0.017$ for plant height). The same trend was observed for leaf area ($P < 0.001$), while stomata density increased with elevation ($P = 0.005$). SLA values for this species were not significantly correlated with elevation ($P = 0.114$) (Fig. 5).

Plant height and leaf area significantly decreased, while stomata density increased, with increasing elevation in *R. purpurea* ($P = 0.044$, $P = 0.001$ and, $P = 0.002$ for plant height, leaf area and stomata density respectively). However, we did not find a significant relationship of elevation either with leaf length ($P = 0.471$) or with SLA ($P = 0.555$) (Fig. 5). Details on the regression analysis including coefficients and associated 95% confidence intervals are provided in Tables 2 and 3.

## DISCUSSION

### Variations of morphological traits with elevation

In the current study, we found a significant decrease in plant height of both species of *Roscoea* (*R. alpina* and *R. purpurea*) with increased elevation. Reduction of plant height in these alpine gingers with increased elevation is consistent with several previous findings reported for tree species (*Körner, 1998*; *Cordell et al., 1998*; *Kronfus & Havranek, 1999*; *Paulsen, Weber & Korner, 2000*; *Kogami et al., 2001*; *Li, Yang & Kräuchi, 2003*; *Shi et al., 2006*) and herbaceous species (*Takahashi & Matsuki, 2017*; *Kiełtyk, 2018*). Similarly, a

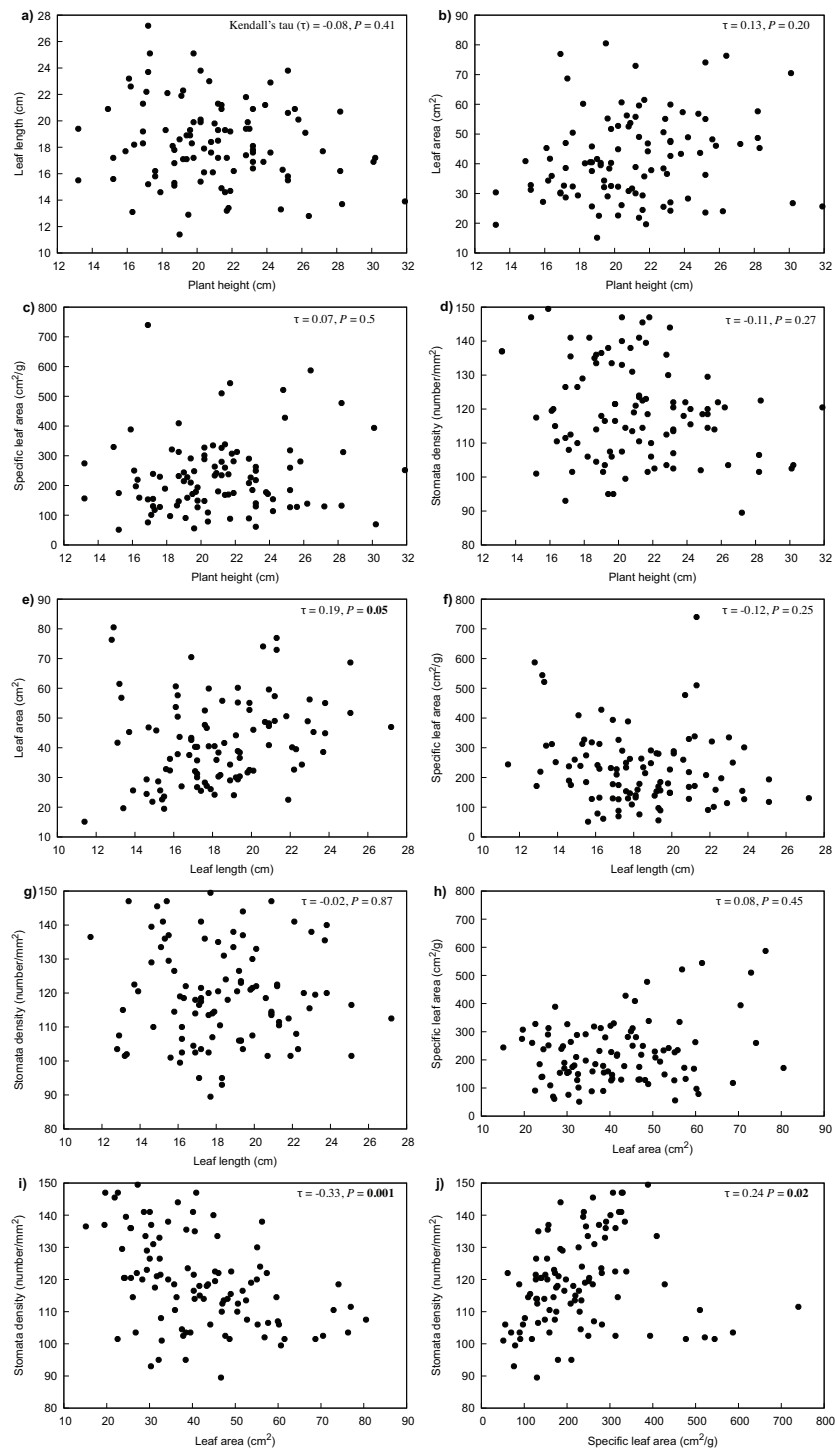

**Figure 3** **Correlations between the different traits of *R. purpurea* measured at five different elevations.**
(A) correlation between plant height and leaf length; (B) correlation between plant height and leaf area;
(C) correlation between plant height and specific leaf area; (D) correlation between plant height and stom-
ata density; (E) correlation between leaf length and leaf area; (F) correlation between leaf length and spe-
cific leaf area; (G) correlation between leaf length and stomata density; (H) correlation between leaf area
and specific leaf area; (I) correlation between leaf area and stomata density; (J) correlation between spe-
cific leaf area and stomata density.

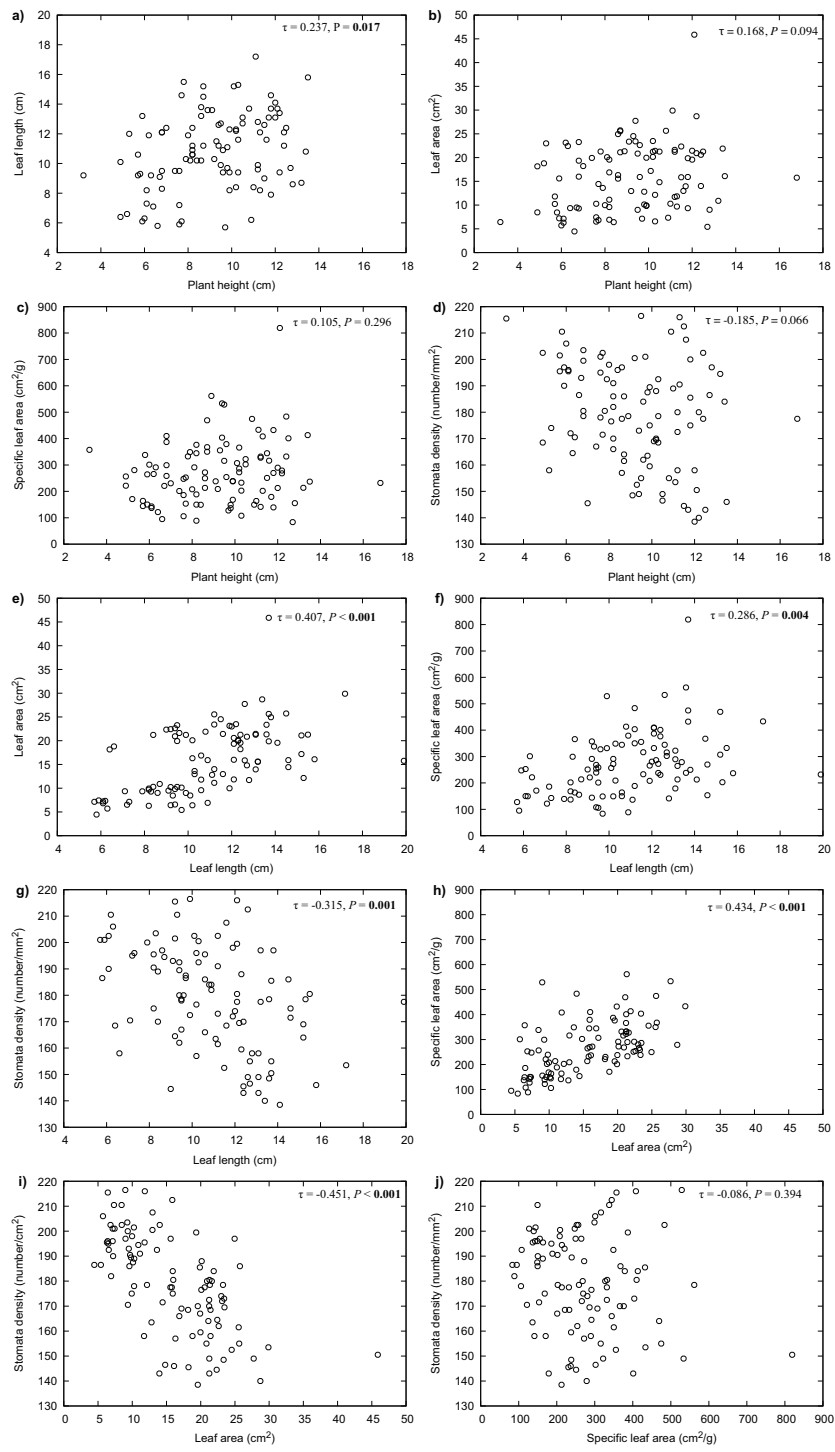

**Figure 4** **Correlations between the different traits of *R. alpina* measured at five different elevations.** (A) correlation between plant height and leaf length; (B) correlation between plant height and leaf area; (C) correlation between plant height and specific leaf area; (D) correlation between plant height and stomata density; (E) correlation between leaf length and leaf area; (F) correlation between leaf length and specific leaf area; (G) correlation between leaf length and stomata density; (H) correlation between leaf area and specific leaf area; (I) correlation between leaf area and stomata density; (J) correlation between specific leaf area and stomata density.

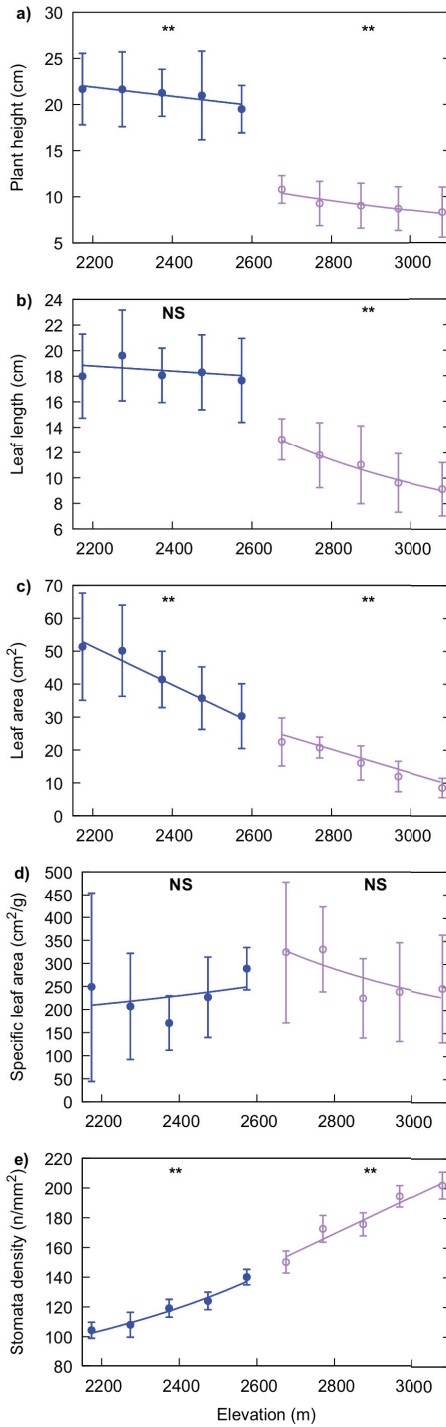

**Figure 5** **Generalised linear regression models showing the effect of elevation on plant height (A), leaf length (B), leaf area (C), specific leaf area (D) and, stomata density (E) for *R. purpurea* (blue line with filled markers) and *R. alpina* (purple line with empty markers).** Markers indicate the mean value of the corresponding trait at each elevation and error bars indicate standard deviation. Solid lines represent the regression function for each trait and species. A significant correlation of elevation on the value for each trait is indicated by two asterisks (**) while a non-significant correlation of elevation is indicated by "NS" above the corresponding regression line.

**Table 2** Results of regression analysis between various traits of *R. alpina* and elevation.

| Traits | Parameters | Coefficients and 95% Cis | | | Distribution | link | P |
|---|---|---|---|---|---|---|---|
| | | 2.5 | 50 | 97.5 | | | |
| Plant height | m | 3.85E−05 | 6.48E−05 | 9.12E−05 | Gamma | Inverse | |
| | b | −1.52E−01 | −7.71E−02 | −1.85E−05 | | | 0.017 |
| Leaf length | m | 6.99E−05 | 8.37E−05 | 9.75E−05 | Gamma | Inverse | |
| | b | −1.87E−01 | −1.47E−01 | −1.08E−01 | | | **<0.001** |
| Leaf area | m | −4.00E−02 | −3.60E−02 | −3.30E−02 | Gamma | Identity | |
| | b | 1.09E+02 | 1.21E+02 | 1.33E+02 | | | **<0.001** |
| SLA | m | 3.91E−07 | 3.33E-06 | 6.31E−06 | Gamma | Inverse | |
| | b | −1.43E−02 | −5.85E−03 | 2.55E−03 | | | 0.114 |
| Stomata Density | m | 9.10E−02 | 1.24E−01 | 1.58E−01 | Gaussian | Identity | |
| | b | −2.75E+02 | −1.78E+02 | −8.15E+01 | | | 0.005 |

Notes.
SLA, specific leaf area.

**Table 3** Results of regression analysis between various traits of *R. purpurea* and elevation.

| Traits | Parameters | Coefficients and 95% Cis | | | Distribution | link | P |
|---|---|---|---|---|---|---|---|
| | | 2.5 | 50 | 97.5 | | | |
| Plant height | m | −7.90E−03 | −5.00E−03 | −2.10E−03 | Gaussian | Identity | 0.044 |
| | b | 2.59E+01 | 3.29E+01 | 3.99E+01 | | | |
| Leaf length | m | −7.00E−03 | −2.00E−03 | 3.00E−03 | Gamma | Identity | 0.47 |
| | b | 1.15E+01 | 2.32E+01 | 3.49E+01 | | | |
| Leaf area | m | −6.70E−02 | −5.80E−02 | 1.55E+02 | Gamma | Identity | **<0.001** |
| | b | −4.80E−02 | 1.79E+02 | 2.03E+02 | | | |
| SLA | m | −7.55E−06 | −1.90E−06 | 3.71E−06 | Gamma | Inverse | 0.55 |
| | b | −4.40E−03 | 8.91E−03 | 2.26E−02 | | | |
| Stomata density | m | −7.31E−02 | −6.18E−06 | −5.05E−06 | Gamma | Inverse | 0.002 |
| | b | 2.04E−02 | 2.32E−02 | 2.59E−02 | | | |

Notes.
SLA, specific leaf area.

decrease of leaf length of *R. alpina* with the increased elevation in the current study is consistent with the previous findings (*Hansen-Bristow, 1986*; *Schoettle, 1990*; *Kajimoto, 1993*; *Kao & Chang, 2001*; *Kiełtyk, 2018*). Based on the present result, we conclude that the elevational gradient has a significant effect on the growth form of these alpine gingers. At the lower elevation, environmental conditions are likely to be more favourable for optimum plant growth. The reduction of plant height and leaf length of these two alpine gingers with increasing elevation reflects the morphological adaptation to increased environmental stresses such as low concentration of carbon dioxide, decreased temperature, higher solar radiation and/or low water availability (*Wang & Gao, 2004*; *Davis, Shaw & Etterson, 2005*; *Guerin, Wen & Lowe, 2012*). The observed relatively smaller plants with shorter leaf characteristics of these gingers at higher elevation thus may reflect local adaptation at a

higher altitude to enable reduction of transpiration and maintain efficient utilization of water (*Ackerly et al., 2002*; *Royer et al., 2008*; *Peppe et al., 2011*; *Guerin, Wen & Lowe, 2012*).

## Variation of physiological traits with elevation
### Leaf traits variation with elevation

Our findings revealed that variation in leaf area showed a significant but negative correlation with elevation, while the correlation between SLA and elevation was non-significant. Consistent with our result, *Kouwenberg, Kurschner & McElwain (2007)* found a decreasing trend in the leaf area of *Quercus kelloggii* with increasing elevation. Our result on the variation of leaf characters (leaf area and SLA) with the elevation is partially consistent with the previous findings reported by *Hultine & Marshall (2000)*; *Scheepens, Frei & Stöcklin (2010)*; *Hulshof et al. (2013)*; *Bastida, Rey & Alcántara (2015)*, while the findings of *Gonzalo-Turpin & Hazard (2009)* indicate a different effect. Previous studies have suggested that the environment at higher elevations is characterized by higher solar radiation, lower water availability and lower stomatal conductance (*Parkhurst & Loucks, 1972*; *Givnish & Vermeij, 1976*; *Ackerly et al., 2002*). Under such potentially stressful environmental conditions, small leaf size provides optimum adaptation to the plants by reducing boundary layer resistance and maintaining favorable leaf temperature and high photosynthetic water use efficiency (*Renzhong et al., 2001*). Thus, decreased leaf area of these alpine gingers with increased elevation may reflect an adaptation for the increased environmental stress and may be favourable to reduce water loss and maintain efficient use of absorbed water (*Renzhong et al., 2001*). In addition, some authors have implicated increasing UVB radiation levels as having a damaging effect on certain plant structures (*Jansen, Gaba & Greenberg, 1998*; *Rozema, Aerts & Cornelissen, 2002*); and there is some evidence of this affecting plant growth in some lowland terrestrial species (*Rozema, Aerts & Cornelissen, 2002*). These topics may be of high value to explore in alpine environments where there are likely large changes in UV levels. SLA is closely associated with leaf thickness, which mediates the trade-off between light capture, water loss and diffusion of carbon dioxide (*Oberle & Schaal, 2011*). Higher SLA leaves are thicker and contain more photosynthetic enzymes and there is more demand for carbon dioxide per unit area. Thus, stomata density increases to supply the higher demand for carbon dioxide. Consequently, the increase in SLA may be an advantage for carbon dioxide uptake. Non-linear change of SLA of both species along the elevational gradient may indicate that environmental factors associated with altitude alone cannot regulate the trade –off between light capture, water loss and diffusion of carbon dioxide in these alpine gingers. The smallest SLA of *R. alpina* at 2,674 m a.s.l and *R. purpurea* at 2,374 m a.s.l may indicate limited carbon gain and supply due to poor availability of resources and may be associated with the least productive zone of these species where retention of captured resources and protection from desiccation is of high priority (*Wilson, Thompson & Hodgson, 1999*).

### Variation in Stomata density with Elevation

We found a significant increase in stomata density of both species (*R. alpina* and *R. purpurea*) with increased elevation. Many authors have made comprehensive efforts to relate the variation in stomata density along elevation gradients

and have obtained different results. Consistent to our current results, *Körner & Cochrane (1985)*, *Friend & Woodward (1990)*, *Hovenden & Brodribb (2000)*, and *Kouwenberg, Kurschner & McElwain (2007)* have found that stomata density increased linearly with elevation. *Li et al. (2006)* found that stomata density of *Quercus aquifolioides* increased linearly up to the height of 2,800 m a.s.l., whilst above that height, it decreased linearly. *Schoettle & Rochelle (2000)* found that the stomata density of *Pinus flexilis* decreased linearly with altitude whilst *Woodward (1986)* did not observe any significant change in stomata density of *Vaccinium myrtillis* considering altitudes from 200 to 1,100 m asl. The significant increase in stomata density with increasing elevation in our findings may be associated with lower availability of carbon dioxide, higher UV-B and long wave radiation, all reducing photosynthetic efficiency by decreasing stomatal absorption and conductance (*Kouwenberg, Kurschner & McElwain, 2007*; *Körner, 2007*). To adapt to such a harsh environmental conditions and maintain vitalities, stomata density of these gingers may have increased. The increase in stomata density provides compensation against the reduced stomatal conductance and carbon dioxide partial pressure to maintain photosynthetic efficiency (*Kao & Chang, 2001*; *Kouwenberg, Kurschner & McElwain, 2007*; *Körner, 2007*).

Our results indicate two major patterns in the vegetative traits of these alpine gingers with increased elevation: a significant decrease of leaf area and a significant increase of stomata density. These variations provide compensation to cope with the change in the concentration of atmospheric carbon dioxide, temperature, humidity and light at higher altitudes (*Van de Water, Leavitt & Betancourt, 1994*; *Hultine & Marshall, 2000*; *Qiang et al., 2003*). A non-significant correlation between SLA and stomata density may suggest that leaf thickness have little role in regulating the carbon dioxide uptake and transpiration in these two alpine gingers. A negative correlation of stomata density with leaf area has previously indicated that with the increase of stomata density at a higher elevation, narrowing of leaves may reduce excess transpiration (*Herms & Mattson, 1992*; *Etterson & Shaw, 2001*). The closely correlated variation in these two traits thus maintains a likely trade-off between photosynthesis and transpiration and provides local adaptation to the specific conditions, at different elevations.

## CONCLUSIONS

Growth, morphology and physiology of *R. alpina* and *R. purpurea* were found to have a significant association with altitude. These alpine gingers exhibit optimum growth at their respective lowermost distribution range, and their growth response retards with increasing elevation. Based on the present result, it can be concluded that these alpine gingers favour shorter height, smaller leaf and higher stomata density at a higher elevation to adapt with the stressful factors associated with the change in elevational gradients. Variation in those traits at different elevations may reflect the response to the combined selection pressure of different abiotic and biotic factors that may generate different micro-environmental conditions at the respective elevation. Decreased growth forms and leaf area of these alpine gingers at a higher altitude may indicate a selection response to reduce water loss

from the plant body during transpiration while increased stomata density may indicate the adaptation to cope with the decreased concentration of carbon dioxide. The closely correlated modification of these traits at different elevations may have played a significant role in providing local adaptation to these alpine gingers.

## ACKNOWLEDGEMENTS

We are thankful to the Department of Plant Resources, Thapathali, Kathmandu for providing research permission. We thank Mr Kul Prasad Lamichhane, Kapil Paudel, and Dipesh Baral for assistance in the field. We thank Dr Subodh Adhikari for the constructive comments on the earlier version of the manuscript. We thank Dr Lalina Muir for proof reading the manuscript. We also thank Editor Dr Gabriele Casazza, reviewer Dr Marco Porceddu and an anonymous reviewer for their constructive suggestion.

### Funding
This work was supported by the Australian Research Council Discovery Projects funding scheme DP160100161 to Adrian G Dyer. The funders had no role in study design, data collection and analysis, decision to publish, or preparation of the manuscript.

### Grant Disclosures
The following grant information was disclosed by the authors:
Australian Research Council Discovery Projects funding scheme: DP160100161.

### Competing Interests
The authors declare there are no competing interests.

### Author Contributions
- Babu Ram Paudel conceived and designed the experiments, performed the experiments, analyzed the data, contributed reagents/materials/analysis tools, prepared figures and/or tables, authored or reviewed drafts of the paper, approved the final draft.
- Adrian G Dyer and Mani Shrestha analyzed the data, contributed reagents/materials/-analysis tools, authored or reviewed drafts of the paper, approved the final draft.
- Jair E Garcia analyzed the data, contributed reagents/materials/analysis tools, prepared figures and/or tables, authored or reviewed drafts of the paper, approved the final draft.

### Field Study Permissions
The following information was supplied relating to field study approvals (i.e., approving body and any reference numbers):
The Department of Plant Resources, Thapathali, Kathmandu provided research permission (125/05-19).

### Data Availability
The following information was supplied regarding data availability: Raw data is available as a Supplemental File.

## Supplemental Information

Supplemental information for this article can be found online at http://dx.doi.org/10.7717/peerj.7503#supplemental-information.

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
