# Peer review of "The effect of elevational gradient on alpine gingers (Roscoea alpina and R. purpurea) in the Himalayas"

_PeerJ, doi:10.7717/peerj.7503_

## Round 0.1 · original submission · Major Revisions

Thank you very much for your submission to PeerJ. Manuscript entitled "How does the elevational climate affect alpine gingers (Roscoea alpina and R. purpurea) in the Nepalese Himalayas?" which you submitted to the journal, has been kindly reviewed by two reviewers. Please answer all the he comments of the reviewers. Reviewers suggest that in general your findings are relevant and meaningful. However, one of the reviewers suggested that putting the job in a wider context will give more weight to the manuscript. So, we encourage you to improve the manuscript according to tips of reviewers and to carefully respond to the comments of reviewers.

Once again, thank you for submitting your manuscript to PeerJ and we look forward to receiving your revision.

Sincerely,
Gabriele Casazza

Reviewer 1 ·

Basic reporting

The article would benefit from professional English correction made by native speaker familiar with the research area.

Literature references are not sufficient in Introduction and Discussion sections. Citing papers dealing with herbaceous plants performance along an elevational gradient are required. Some references are suggested in General comments section.

The MS has professional article structure, figures and tables. Raw data is shared.

The MS is self-contained and include all results relevant to the hypothesis.

Experimental design

The MS presents primary research within aims and scope of the journal.

Research question is well defined, relevant and meaningful. However, Authors have not properly identified knowledge gap being investigated, because literature review they have made is not sufficient. According to Authors aim of the study is ‘to provide the primary information on how alpine herbs do respond with the potentially stressful environmental conditions associated with increasing elevation in Himalayas.’ (Lines 73-75). The most important question should be how do alpine herbs respond along an elevational gradient in mountains, not only in some particular region of Himalayan Mts. Therefore, aims of the MS should be placed in the wider context of available literature dealing with herbaceous plants responses along elevational gradients, also from other high mountain regions of the world.

The research was rigorously performed to a high technical and ethical standards.

Methods are described with sufficient detail and information to replicate.

Validity of the findings

The MS represents important and valuable contribution to knowledge on herbaceous plants responses to environmental conditions along an elevational gradients in mountains. Papers dealing with this topic are not numerous and there is urgent need to gather more evidences on how plants respond along elevational gradients. In the context of predicted future climate changes, knowledge of plant performance along elevational gradients may contribute to an enhanced prediction of plant responses under an altered climate.

Data is robust and properly analysed by statistical methods.

Conclusions are well stated and linked to original research question. In the General comments section I commented on unsupported by results discussion of optimum photosynthetic zone (Lines 251-253) and unsupported conclusion on optimum growth zones (Line 292-293). Other conclusions are supported by results.

Speculations are identified as such.

Additional comments

I found your manuscript to be interesting and valuable contribution to the research area. Beside the other investigated morphological traits, analysis of stomata density variation along an elevational gradient is particularly interesting. Here, I present detailed comments to the MS, that I believe may help you to improve the article. Because I am not native English speaker, please, treat all language corrections I have made as suggestions.

Title: How does the elevational climate affect ...
I suggest to change this title. I am not sure if there is something what can be called ‘elevational climate’. Climate parameters change along an altitudinal (elevational) gradient (elevational shift in climate), but there is no common ‘elevational climate’. Moreover, are Authors sure that only climate affect morphology of the studied plants along elevational gradient?
I suggest something like this: How does an elevational gradient affect... or Variation of ...along elevational gradient

Abstract
Line 26: ‘the elevational gradient’
Perhaps here should be ‘an elevational gradient’

Lines 28-29: ‘Despite this interest...’
I suggest to delete this sentence. I agree that field works in remote high mountains is a challenging task, however, it should not be mentioned in Abstract as rationale for publishing results of your research.

Line 32: after meters ‘m’ add ‘a.s.l.’

Line 32-33: ‘We measured...at five plots, along the elevational transect at an elevational gap of ca. 100 m’
This description is somewhat not clear. I suggest to write that both species were investigated at five plots distributed along two vertical transects of 400 m, R. purpurea at elevational range of 2174-2574 m a.s.l. in Daman (Mt.?), and R. alpina at 2675-3079 m in Ghorepani (Mt. ?).

Lines 37-39: ‘These findings provide (a) an insight ....(b) provide relevant information...’
This sentence is vague; what are these ‘insight’ and ‘relevant information’? I suggest to present some of your conclusion here.
And again, think of replacing phrase ‘elevational climate’

Line 41: ‘Key words: Climate change’
I suggest to delete ‘climate change’ here because in the MS there are no discussion on possible effects of climate change on the studied species.


Introduction
Lines 49-50: ‘...small changes in elevation can lead to large variation in temperature...’
I suppose it should be: ‘can lead to large shift in temperature’

Line 70: ‘along altitudinal gradient’
Please use the terms altitude and elevation consistently (not interchangeably) throughout the MS (see McVicar TR, Körner Ch (2013) On the use of elevation, altitude, and height in the ecological and climatological literature. Oecologia 171:335–337)
In most cases in the MS ‘elevational gradient’ or ‘elevation’ is appropriate, but when Authors refer to climatic or environmental conditions such as, for example, temperature or CO2 pressure then altitude or altitudinal gradient is adequate.

Lines 69-71: ‘Several studies on the growth and physiological responses of trees along altitudinal gradient...'
Why did Authors cite only literature concerning trees? There are also available studies on performance of herbaceous plants along an elevational gradient in mountains. I suggest to provide here an outline of morphological variation in vegetative and generative traits of herbaceous plants. Here I recommend some papers:
Kiełtyk P (2018) Variation of vegetative and floral traits in the alpine plant Solidago minuta: evidence for local optimum along an elevational gradient. Alp Bot 128:47–57
Seguí J et al (2018) Phenotypic and reproductive responses of an Andean violet to environmental variation across an elevational gradient. Alp Bot 128:59–69
Bastida JM et al (2015) Local adaptation to distinct elevational cores contributes to current elevational divergence of two Aquilegia vulgaris subspecies. J Plant Ecol 8:273–283
Maad J et al (2013) Floral size variation in Campanula rotundifolia (Campanulaceae) along altitudinal gradients: patterns and possible selective mechanisms. Nord J Bot 31:361–371
Hulshof CM et al (2013) Intra-specific and inter-specific variation in specific leaf area reveal the importance of abiotic and biotic drivers of species diversity across elevation and latitude. J Veg Sci 24:921–931
Alexander JM et al (2009) Establishment of parallel clines in traits of native and introduced forbs. Ecology 90:612–622
Gonzalo-Turpin H, Hazard L (2009) Local adaptation occurs along altitudinal gradient despite the existence of gene flow in the alpine plant species Festuca eskia. J Ecol 97:742–751
von Arx G et al (2006) Evidence for life history changes in high-altitude populations of three perennial forbs. Ecology 87:665–674

Lines 72-75: ‘...there is a paucity of available information...’. ‘...this study aims to provide the primary information...’

There are some studies on performance of alpine herbaceous plants along elevational gradient and you should cite some of them. This study do not provide the primary information. But, your study makes an important contribution to this research area and, especially, stomata density has been rarely investigated along elevational gradient in herbaceous plants.

Line 74: ‘respond with’
- ‘respond to’

Lines 76-82: ‘The genus Roscoea, with 22 known species, is a Himalayan endemic alpine perennial herb...’. ‘Among the currently known seven Himalayan Roscoea species...’
Please, clarify here the number of Roscoea species: 22 or 7? These two sentences seem to be in contradiction.

Lines 87-89: ‘...we aimed to explore how do morphological (plant height and leaf length) and physiological variables (leaf area, SLA and stomata density)...’
and Lines 135-136: ‘Plant height and leaf length were measured to examine the morphological variables. Physiological variables included leaf area, specific leaf area (SLA) and stomata density.’

I am not sure if it is correct to classify leaf area and stomata density as physiological variables. I think these variables are morphological traits (stomata density may be regarded as micro-morphological trait). However, I agree that these traits are linked and related to physiological processes and can be used to assess and discuss plant physiology. Please, check classification of these traits in literature.

Materials and Methods
Study species
Lines 94-95: ‘Roscoea alpina is a common species with a wide distribution between the elevations of 2130-4270 m a.s.l. ...’
Lines 107-108: ‘Roscoea purpurea is the other most widespread members of the Himalayan Roscoea, distributed between the elevations of 1520-3100 m a.s.l. ...’

R. alpina was sampled from elevational range 2675-3069 m a.s.l. (404 m vertical range) what makes only 19% of the elevational range of this species. R. purpurea was sampled from elevational range 2174-2574 m a.s.l. (400 m vertical range) what makes 25% of the elevational range of this species. Therefore, both species were investigated only in central parts of their elevational distribution in Himalayan Mountains. Authors should be aware of this when interpreting pattern of elevational variation in the investigated species. If there are some reasons why these species were sampled only from 400 m vertical range explain it in Material and Methods section. And I suggest to give some restriction concerning this issue at the end of Discussion section.

Line 129: ‘We present the geographical details of the study sites in Table 1’
I suggest: Geographic coordinates and elevations of the study sites are presented in Table 1.

Line 135: ‘cc 100 m’
- ca. 100 m

Line 152: ‘...to access the variability.’
- to assess variability

Line 182: ‘... with low N.’
- with low sample size.

Results
Line 196: ‘Effect of altitude on physical traits ...’
- ‘physical’? Here it would be adequate the title from Line 209: ‘Variation of...’

Line 200: ‘...were not significantly affected by elevation...’
- were not significantly correlated with elevation

Line 203: ‘...we found no significant effect of elevation on either leaf length...’
- Suggestion: ...we found no significant relationship between elevation and leaf length... , as well as SLA’.
Generally, I suggest to avoid such definite statement as ‘effect’, ‘affected’ unless you have evidences on causal relationship.

Discussion
Line 210-214
Please, refer to some papers on elevational variation in morphology of herbaceous plants.

Line 219: ‘... as evidenced by the wider abundance of Zingiberaceae...’
- In my opinion this is not compelling argument.

Line 225: ‘...thus reflects...’
- thus may reflect

Line 249-: ‘Non-linear change of SLA of both species along the elevational gradient may indicate...’
In Results Authors stated: ‘we found no significant effect of elevation on ... SLA’ (Lines 203-204). Lack of significant effect in linear models do mean that there is significant non-linear change. On Fig 5 Authors presented mean values accompanied with +/- 1 standard deviation bars. The observed value can fall everywhere within range +/- 1sd with probability of 0.68. Because ranges of +/- 1sd for most of elevational samples for SLA overlapped considerably, what means high variation in this trait, and we cannot conclude if there is any significant change across elevation or merely random variation. Therefore, discussion of possible optimum photosynthetic zones (Lines 251-256) is not supported by the presented data. Moreover, analysed data was collected from 0.19-0.25 central fraction of elevational ranges of both species and may represent entirely optimal zones.

Line 261: ‘...have obtained mixed results.’
- different/contrasting results?

Lines276: ‘Our results indicate two major variations in ...’
- two major patterns in...?

Conclusions
Lines 292-293: ‘Thus, optimum growth zone of ...’
Results of this study (400 m vertical gradient which makes 0.2-0.25 fraction of elevational distributions of both species) do not support discussion of detailed elevations of optimum zones.

Figure 2: Please, indicate where did you collect R. alpina and R. purpurea.

Figure 3 and Figure 4: I suggest to put statistics directly on plots (at upper-right corners for example). Significant statistics may be distinguished with bold type.

Figure 5. Write (a), ... (b)... instead of (panel a), ... (panel b)

Raw data (csv file):
- add units to varaibles
- describe clearly which variables are for R. purpurea and which for R. alpina (Plant_height and Plant_height1 are not clear)

·

Basic reporting

no comment

Experimental design

no comment

Validity of the findings

no comment

Additional comments

In my personal opinion, the study and the topics presented in this paper are very interesting, and the paper is enjoyable to read. The experiments appear well designed and the data properly analysed. I think that the results reported in the manuscript have important scientific significance.

I work in particular with seeds, so I would be curious to know if some variation is also reflected in the seeds, so I suggest to the authors (in the future) to add also the seed traits in their analysis.

I give some suggestions/comments:
I suggest to report the elevation in “m a.s.l.” instead to “m” along all the text.

In LL 93 and 211 “and” no italics.

LL 152-154 and LL 174-175: I suggest to move these information to the Results section.

About the Results section, I was surprised to see the text so short, but in accord with the choice of the authors, I think that the figures and the tables summarize in a pleasant way the great amount of results.

Review the formatting of table 2, in particular the column reporting the p values.

---

## Round 0.2 · Minor Revisions

All changes solicited from the reviewer are done and according to their opinion your paper "Vegetation dynamics of abandoned paddy fields and surrounding wetlands in the lower Tumen River Basin, Northeast China " was significantly improved. Nevertheless, a reviewer suggests a few minor changes to further improve the text.

Therefore, I suggest to make these small changes before manuscript acceptance.

Once again, thank you for submitting your manuscript to PeerJ and we look forward to receiving your revision as soon as possible.

Sincerely,
Gabriele Casazza

Reviewer 1 ·

Basic reporting

no comment

Experimental design

no comment

Validity of the findings

no comment

Additional comments

The submitted version of the MS is thoroughly revised and significantly improved. Authors used clear and unambiguous, professional English throughout. In their rebuttal letter authors have given satisfactory responses to all my comments and suggestions. I recommend to accept the submitted manuscript for publication in PeerJ journal without need for additional revision.

·

Basic reporting

no comment

Experimental design

no comment

Validity of the findings

no comment

Additional comments

Dear Authors,

I thank you for taking my previous pieces of advice into consideration. I read with attention the final version and, for me, this version is improved respect the previous.

In my personal opinion, the paper now is OK and I suggest only other very small integration/corrections:

LL 96: Royle and Smith, no italics.

LL 135-136 and after in L 140: When you report “…vegetation type … of Pinus, Rhododendron and Quercus”, is it possible to specify better the species of Pinus, Rhododendron and Quercus present? The complete information could help the readers in better understanding the vegetation of the sites without further analysis "outside" this paper.

L 146: 2574 m a.s.l. and 3079 m a.s.l.

Table 1: I suggest to complete the coordinate details adding “N” in latitude columns, and “E” in longitude columns.

Table 2 and Table 3: I suggest to explain in full SLA in the tables or, probably, you could simply report in the caption “SLA correspond to specific leaf area" or something similar.

---

## Round 0.3 · accepted · Accept

All changes solicited from the reviewer are done. So, I am very pleased to say that your paper "The effect of elevational gradient on alpine gingers (Roscoea alpina and R. purpurea) in the Himalayas " is accepted for publication in the PeerJ.

Best regards
Gabriele Casazza